# Smoothness and Efficiency Metrics Behavior after an Upper Extremity Training with Robic Humanoid Robot in Paediatric Spinal Cord Injured Patients

Miriam Salas-Monedero [1,2], Víctor Cereijo-Herranz [3], Ana DelosReyes-Guzmán [1,4,*], Yolanda Pérez-Borrego [5], Angel Gil-Agudo [4,6], Fuensanta García-Martín [3], José-Carlos Pulido-Pascual [3] and Elisa López-Dolado [6]

1   Biomechanics and Technical Aids Unit, Hospital Nacional de Parapléjicos (SESCAM), Finca La Peraleda, s/n, 45071 Toledo, Spain; miriam.salas@alu.uclm.es
2   International Doctoral School, Castilla La-Mancha University, 02071 Albacete, Spain
3   Inrobics Social Robotics, S.L.L., Av. Gregorio Peces Barba, 1, Leganés, 28919 Madrid, Spain; vcereijo@inrobics.com (V.C.-H.); fgarcia@inrobics.com (F.G.-M.); jcpulido@inrobics.com (J.-C.P.-P.)
4   Unidad de Neurorrehabilitación, Biomecánica y Función Sensitivo-Motora (HNP-SESCAM), Unidad Asociada de I+D+I al CSIC, 45071 Toledo, Spain; amgila@sescam.jccm.es
5   Functional Exploration and Neuromodulation of Nervous System Investigation Group, Hospital Nacional de Parapléjicos (SESCAM), 45071 Toledo, Spain; yaperez@sescam.jccm.es
6   Rehabilitation Department, Hospital Nacional de Parapléjicos (SESCAM), Finca La Peraleda, s/n, 45071 Toledo, Spain; elopez@sescam.jccm.es
*   Correspondence: adlos@sescam.jccm.es; Tel.: +34-925-247771

**Abstract:** The upper extremity behavior in smoothness and efficiency metrics should be different between paraplegic and tetraplegic patients. The aim of this article was to analyze the behavior of these metrics after receiving upper extremity training with the humanoid robot Robic as a treatment. Ten pediatric patients participated in the study and completed ten experimental sessions with Robic. Patients were assessed at baseline and at ending the training using the Box and Block test and a non-immersive virtual application based on the Leap Motion Controller available in the RehabHand software. From this application, the smoothness metric was calculated as the number of peaks or units of movement detected in the velocity profile of the hand during the execution of the task, and the efficiency metric was assessed by calculating the length of the hand trajectory. Patients with tetraplegia had a significantly longer trajectory (286.01 ± 59.87 mm) than paraplegics (123.61 ± 17.14 mm) in the baseline situation. However, at the end of the training, there were no differences between them. In the Box and Block test, the paraplegic group passed more cubes than tetraplegics. In conclusion, the first experience with a Robic robot in SCI was very positive, with observed improvements in upper extremity dexterity in trained patients.

**Keywords:** pediatric spinal cord injury; robotic-based rehabilitation therapies; upper extremity training; biomechanics; smooth movement; efficiency metric

## 1. Introduction

The incidence rate of pediatric-onset spinal cord injury (SCI) was estimated to be 3.3 and 6.2 cases per million per year for traumatic and non-traumatic SCI, respectively, in Europe [1]. Among other sequelae, SCI impairs the function of the upper extremity (UE), affecting the patient's level of independence and influencing the performance of activities of daily living and social participation.

After an injury, rehabilitation treatment is approached from different areas and perspectives. Moreover, in the case of the pediatric population, aspects such as adherence and attention during therapeutic sessions are fundamental when analyzing the results of treatment. To increase engagement and quality of life in this population, therapeutic

interventions need to address all factors, including the underlying psychosocial aspects, instead of task and environment modification alone [2].

Advances in Rehabilitation Robotics have introduced new and reliable technologies into the therapeutic process [3]. Clinical experimentation demonstrates that a patient's motivation is a key aspect of the successful implementation of neural rehabilitation therapies with robotic devices [4] and directly correlates to how fully a patient is engaged in the treatment. Another line of research attempts to integrate the motivation factor, including the "social" aspect, into robotic-based rehabilitation therapies [5]. An example would be the Inrobics Rehab Clinic solution, in which Robic (NAO robot, Aldebaran Robotics) provides guided training and social interaction with the patient. NAO robot has been successfully applied in cardiac rehabilitation in a pilot study [6], and the Inrobics Rehab Clinic platform has been previously applied to children with infantile cerebral palsy (CP) and obstetric brachial plexus palsy [7].

However, although all these experiences have been positive, there is no evidence to date of studies that have attempted to quantify the possible improvements in the movement patterns of the UE due to robotic treatment. Movement patterns can be analyzed according to different biomechanical aspects and characteristics of motor control, with the smoothness of movement being one of the most interesting since this is a quality that is highly altered in the presence of neurological pathology [8,9]. Smooth movement is defined as movement towards a target without deviations from its ideal trajectory [8]. Also, the analysis of the efficiency metric is of special interest from the pattern of the executed UE movement based on the evaluation of the trajectory length [10]. Both metrics of UE movement quality, smoothness and efficiency are computed from kinematic data from the hand and are known to be affected in the adult SCI population [9,10] but have not been addressed in children.

The present study is proposed in a pediatric population with SCI and is based on the initial hypothesis that the behavior in these metrics should be different depending on whether the patient suffers from paraplegia or tetraplegia. Therefore, the aim is to analyze the behavior of the smoothness and efficiency metrics after a treatment received with the Robic humanoid robot that proposes the UE training specifically designed in this research for the pediatric population.

The rest of the paper is organized as follows:

- Section 2 describes the methodology in relation to the design and implementation of the UE training by means of the Inrobics Rehab Clinic platform; the characteristics of the participants; the experimental protocol and the content of each experimental session; and the variables analyzed;
- Section 3 presents the results in relation to the variables described at baseline and at ending the UE training;
- Sections 4 and 5 included the discussion and conclusions of the study, respectively.

## 2. Materials and Methods

### 2.1. Study Design and Participants

The present study was approved by the Local Ethics Committee (Comité Ético de Investigación Clínica con Medicamentos, Complejo Hospitalario de Toledo; Approval number: 760 (29 September 2021)).

A prospective observational study was carried out. The participants were enrolled and assigned to the UE training for clinical staff. The recruitment was made on a sample of 10 children with chronic SCI from the population treated at Hospital Nacional de Parapléjicos (Toledo, Spain). The inclusion criteria were: (1) to have an SCI of at least C6 metameric level in the case of complete motor injuries, classified according to the International Standards for the Neurological Classification of Spinal Cord Injury (ASIA Impairment Scale (AIS)) [11] as A or B AIS grades, or any level of incomplete AIS C or D SCI that allows us to perform UE reaching movements; (2) age between 7 and 14 years old; (3) to have reached the seated posture and to sign the corresponding written informed consent. The exclusion criteria were: to have an unstable orthopedic injury such as un-

consolidated fractures or with unstable osteosynthesis systems in UE; skin lesions and/or pressure ulcers in the exoskeleton placement area; joint stiffness and/or severe spasticity; bronchopneumopathy and/or severe heart disease that will require monitoring during the exercise; visual problems and cognitive impairment. The dominant hand of the patients was taken into account and assessed by means of the Edinburg Handedness Inventory [12]. The Upper Extremity Motor Score (UEMS) was obtained, with the clinical staff assessment of the strength of 5 muscle groups of the UE. Each muscle group can be evaluated between 0 (no function) and 5 (normal function), with a total of 25 points for each UE. All patients/parents signed an informed consent form before the study.

The guidelines of the Declaration of Helsinki were followed in every case. Subject demographics are provided in Table 1. No patients who did not meet the inclusion criteria were selected, and, therefore, none were excluded either.

**Table 1.** Demographics and functional characteristics of the sample were analyzed.

| Variables | Sample Analyzed | |
|---|---|---|
| | Tetraplegic Patients (n = 5) | Paraplegic Patients (n = 5) |
| Sex (Male) * | 1.00 ± 20.00 | 3.00 ± 60.00 |
| Age (Years) + | 9.67 ± 4.04 | 10.67 ± 2.08 |
| Weight (kg) | 39.66 ± 7.50 | 36.03 ± 12.22 |
| Height (cm) | 138.66 ± 11.06 | 140.33 ± 16.92 |
| Etiology Injury (Traumatic) | 1.00 ± 20.00 | 1.00 ± 20.00 |
| Time since injury (months) | 3.66 ± 2.51 | 10.00 ± 3.00 |
| Injury Level | C1: 1.00 ± 20.00<br>C2: 1.00 ± 20.00<br>C5: 1.00 ± 20.00<br>C6: 1.00 ± 20.00<br>C7: 1.00 ± 20.00 | T3: 1.00 ± 20.00<br>T12: 1.00 ± 20.00<br>L1: 1.00 ± 20.00<br>L2: 1.00 ± 20.00<br>L3: 1.00 ± 20.00 |
| AIS Classification<br>A<br>B<br>C<br>D | -<br>3.00 ± 60.00<br>1.00 ± 20.00<br>1.00 ± 20.00 | -<br>-<br>3.00 ± 60.00<br>2.00 ± 40.00 |
| Upper Extremity Motor Score | **31.33 ± 15.27 [a]** | **50.00 ± 0.00 [a]** |

Significant statistically differences are expressed in bold font. [a] ($p < 0.01$); * categorical variables are expressed as frequency and percentage; + continuous variables are expressed as mean and standard deviation.

### 2.2. Upper Extremity Training

The UE training was scheduled for 30 min UE experimental sessions, between 2 to 3 sessions per week (maximum 5 weeks for completing the training scheduled in 10 sessions) by means of the Inrobics platform (Figure 1). The sessions' number performed was registered as a feasibility outcome and compliance.

At the start of the session, the humanoid robot Robic was in a resting position, sitting on the floor. The clinician initialized the tool from the APP and simultaneously configured the relevant rehabilitation parameters (Figure 1). In this way, the robot was started, greeted and stood up. This was followed by the experimental session, which initially consisted of 3 exergames separated by 2 periods of resting.

Patients performed the experimental sessions in their own wheelchairs (Figure 2). The sessions were carried out by the same researcher with the collaboration of Inrobics experts. The course of the sessions consisted of initial contact between the patient and the professional with the platform, the development of the session and the closing.

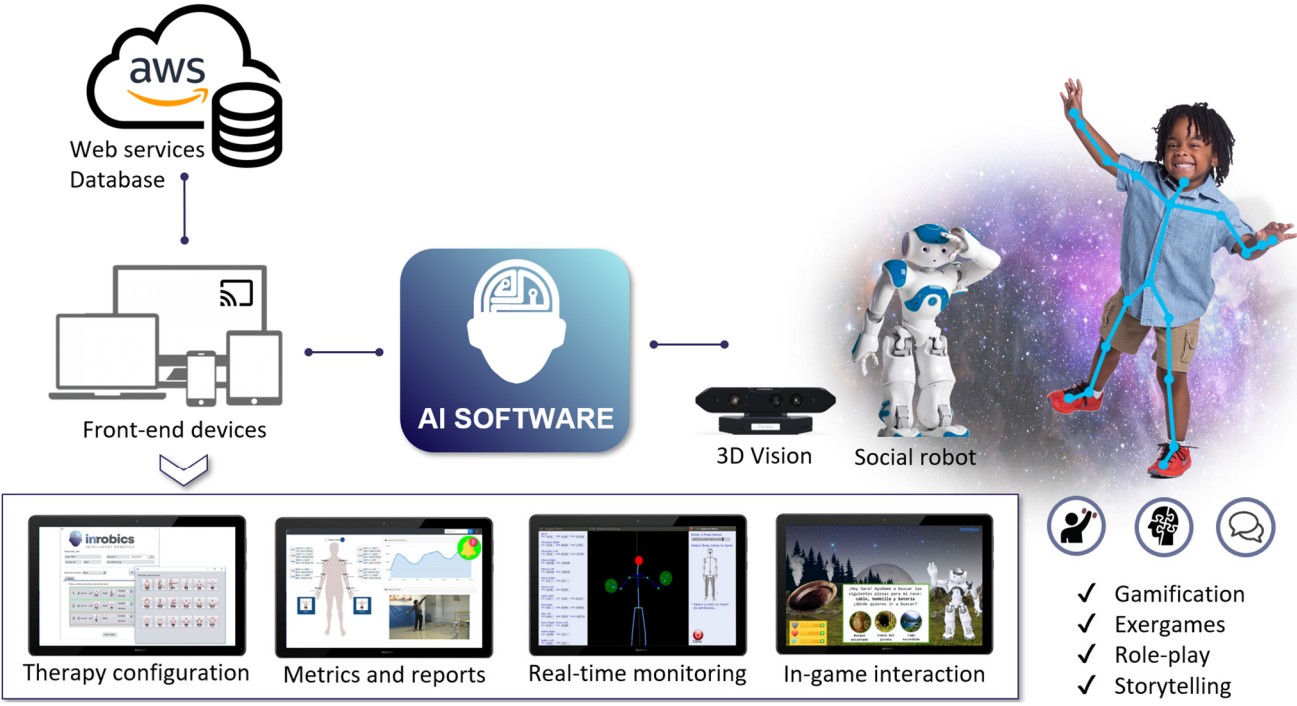

**Figure 1.** Inrobics Rehabilitation Clinical Platform.

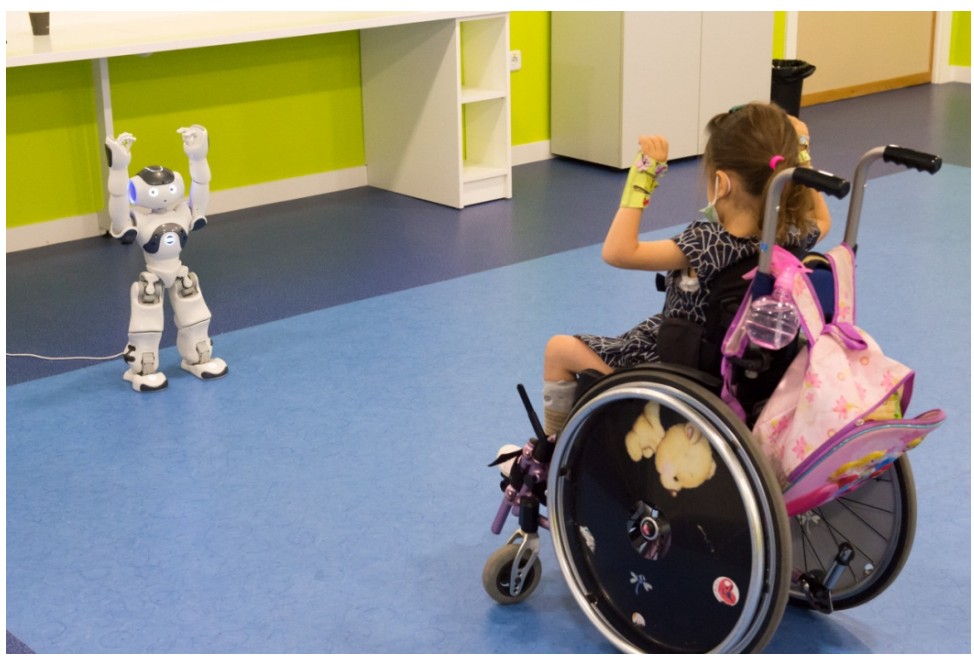

**Figure 2.** Patient during an experimental session with humanoid robot Robic.

The muscle training protocol was developed at the National Paraplegic Hospital in Toledo. The protocol consists of 4 different exercise programs: 2 standing programs and 2 sitting programs. Two were for children (standing and sitting), and the other 2 were for adolescents (standing and sitting). The use of a program for children or a program for adolescents is based on the Tanner scale, considering that the programs for children should be used in stages 1–2 and from stages 3–4 for adolescents. The exercises for each training program are based on the use of the International Classification of Functioning, Disability and Health [13] (ICF) framework. In this framework, a person's disability can be considered in terms of impairments, activity limitations and participation restrictions.

Therefore, the exercises were designed taking into account the limitations and difficulties in order to create exercises and movements suitable for all the proposed tasks and thus to fulfill all the objectives of strength training. For this study, the seated training program for children was chosen, as they all corresponded to level 1. So, each experimental session consisted of performing different exercises from the UE muscle training protocol, with the patient always in a seated position. In Figure 3, the complete content for 1 experimental session is shown in a graphical way.

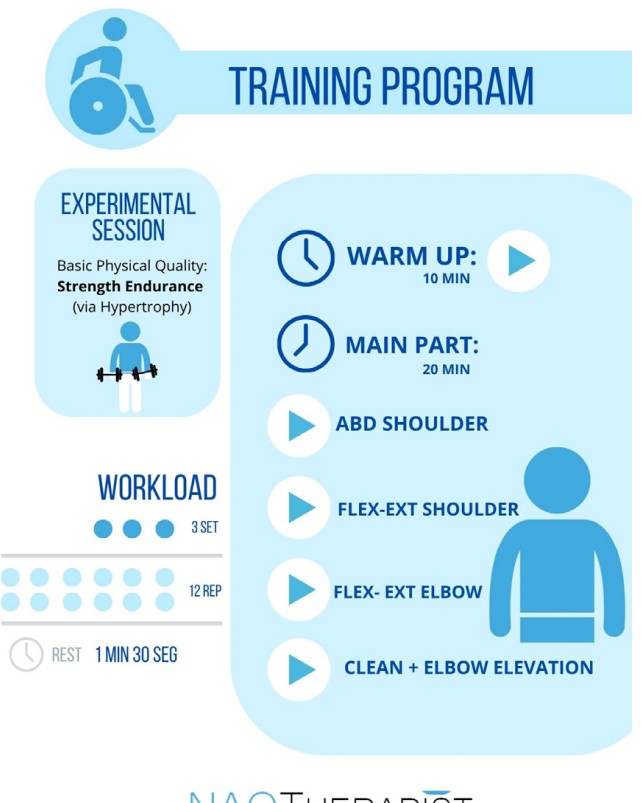

**Figure 3.** Exercises performed during an experimental session with the humanoid robot Robic.

The program consisted of 10 training sessions carried out twice a week, each session lasting approximately 30 min [14–16]. The training session consisted of 3 parts: warm-up, main part and cool-down [17]. The warm-up and cool-down periods lasted approximately 10 min [17], with the warm-up period not differing from the main period in any of the sessions, and only the cool-down period differing. The objectives of the main part of all sessions were based on low-resistance aerobic endurance work. The aim was to perform between 8 and 15 repetitions of the selected muscle groups (deltoids, shoulder rotators, elbow flexors and extensors and wrist flexors and extensors) in a way that was sufficient to meet the strength endurance requirements in preparation for repetitive movements, such as those used in neurorehabilitation to acquire movement patterns for activities of daily living. These patterns involve several muscles in the same movement with the common goal of reducing the energy cost of movement and improving the quality of life of children with disabilities [18]. Depending on the patient's performance, the robot provided real-time feedback. The therapist could visualize the sensor monitoring the patient's movements at any time and intervene during the session if necessary. At the end of the session, the robot provided the patient with feedback on their performance.

### 2.3. Outcome Variables

The outcomes variables were collected at baseline and at ending the UE training using a non-immersive virtual application based on the Leap Motion Controller (LMC) [19] avail-

able in the RehabHand software developed at our center [20], specifically the application Explorer performed with the dominant hand. This application proposed the execution of a functional task based on the tracking and tracing of a previously defined trajectory with an envelope shape, passing only once through each node. In this case, with the aim of comparing the trajectories in different situations or between subjects, the order to reach each node is always the same (Figure 4). The kinematic variables of the hand were recorded by means of LMC and saved in a text file that was subsequently analyzed in Matlab. Thus, the smoothness metric was calculated as the number of peaks or units of movement detected in the velocity profile of the hand during the execution of the task, and the efficiency metric was assessed by calculating the length of the hand trajectory performed. Moreover, patients were evaluated by means of the Box and Block test [21].

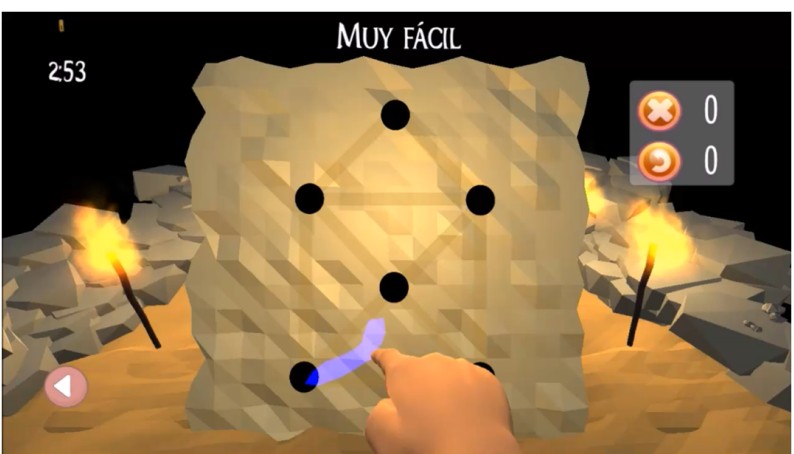

**Figure 4.** Virtual application Explorer included in RehabHand prototype.

*2.4. Data Analysis*

All the statistical analysis was made by the software SPSS 17.0 for Windows (SPSS Inc., Chicago, IL, USA). The clinical and demographic characteristics of the participants were analyzed by descriptive statistics, showing the results as the mean and standard deviation.

The non-parametric Mann–Whitney U-test was applied to analyze the possible differences between both groups analyzed (paraplegic and tetraplegic patients). The variables analyzed with this method were the peak number or movement units, the trajectory length measured in mm and the performance in the Box and Block test measured by the block number. Moreover, these variables were compared at baseline and at ending the complete UE training program for each group of patients by using the Wilcoxon statistical method.

**3. Results**

As a measure of usability and compliance, the number of sessions was recorded for each participant. All completed the UE training with all experimental sessions.

At baseline, no statistically significant differences were found between the two groups of patients analyzed, tetraplegic and paraplegic children, in the demographic and functional variables, with the exception of the UEMS, the result of which differentiated the two experimental groups, obtaining a significantly lower score in the tetraplegic group ($31.33 \pm 15.27$) than in the paraplegic group, which obtained the maximum score ($50.00 \pm 0.00$; Table 1). None of the patients, neither paraplegic nor tetraplegic, showed changes in the assessment of the strength of the main muscles of the upper limb at the end of the exercise, or in other words, there was no change in the UEMS in any of the patients that could be attributed to the robotic training.

The results in terms of path length, number of peaks and Box and Block test for both experimental groups are shown in Table 2.

**Table 2.** Performance in relation to the trajectory, peaks number and the Box and Block test at baseline and at ending the upper extremity training for tetraplegic and paraplegic patients.

| Variables | Tetraplegic Patients (n = 5) | | | Paraplegic Patients (n = 5) | | | p₂ | p₃ |
|---|---|---|---|---|---|---|---|---|
| | At Baseline | At Ending | p₁ | At Baseline | At Ending | p₁ | | |
| Trajectory length (mm) | **286.01 ± 59.87** [a] | 128.73 ± 30.07 | 0.109 | **123.61 ± 17.14** [a] | 114.13 ± 34.59 | 0.285 | **0.004** | 0.700 |
| Peaks number (units) | 81.67 ± 48.21 | 62.67 ± 6.65 | 0.285 | 79.00 ± 31.34 | 44.25 ± 18.57 | 0.285 | 0.700 | 0.700 |
| Box and Block Test (blocks) | **17.83 ± 5.72** [a] | **19.40 ± 5.41** [b] | 0.066 | **39.40 ± 9.02** [a] | **38.75 ± 8.38** [b] | 0.496 | **0.008** | **0.009** |

Significant statistically differences are expressed in bold font. [a,b] ($p < 0.01$); The results are expressed as mean and standard deviation. $p_1$, *p*-value from the Wilcoxon method between the baseline and end conditions; $p_2$ and $p_3$, *p*-value from the Mann–Whitney U method between tetraplegic and paraplegic patients at the baseline and end conditions, respectively.

*Comparison between Groups and between Baseline and Ending Conditions*

The three variables obtained for the two groups were compared at baseline and at ending the UE training.

In the analysis of the baseline condition, statistically significant differences were observed in the measured length of the trajectory between the two selected groups, with the tetraplegic children having a longer trajectory ($286.01 \pm 59.87$) than the paraplegic children ($123.61 \pm 17.14$), p2 = 0.004 (Table 2, Figure 5a). Therefore, in terms of efficiency metric, the performance of the paraplegic group was significantly better.

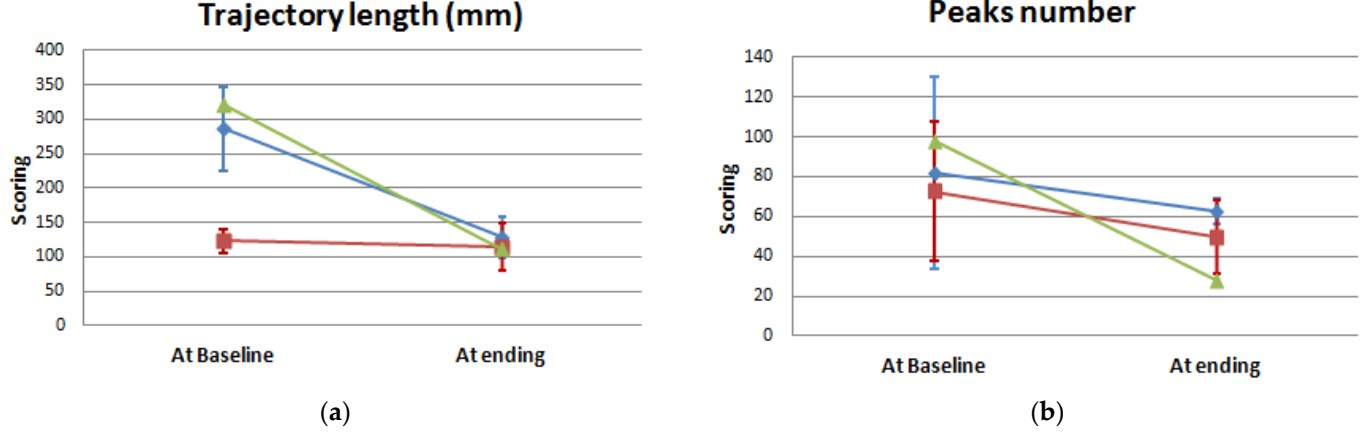

(**a**)                                        (**b**)

**Figure 5.** Graphical results in relation to baseline and end conditions. Tetraplegic group is expressed as blue color, paraplegic as red color and the case of a single patient as green color: (**a**) Results about trajectory length; (**b**) Results about the peaks number.

Regarding the number of peaks, no statistically significant differences were found between the two groups analyzed. The result was slightly higher in the tetraplegic group ($81.67 \pm 48.21$) than in the paraplegic group ($79.00 \pm 31.34$; Table 2, Figure 5b), resulting in a less smooth UE movement.

Regarding the Box and Block test, statistically significant differences were observed between the two groups both at baseline and at the end, with a lower performance in the number of blocks passed in the tetraplegic group (baseline $17.83 \pm 5.72$; ending $19.40 \pm 5.41$) than in the paraplegic group (baseline $39.40 \pm 9.02$; ending $38.75 \pm 8.38$), $p < 0.01$ (Table 2, Figure 6). Interestingly, no significant differences were found in this test performance in relation to the robotic training program, but all tetraplegic children tended to perform better once they had completed the training program that they had before starting it.

For the other two variables, trajectory length and the number of peaks, no significant differences were found between the two experimental groups and between the baseline

and ending conditions (Table 2). This finding can be seen in Figure 6, which shows the overlapping of the results obtained in both tetraplegic and paraplegic patients. However, the improvement experienced by both groups between the two assessments can be seen graphically (Figures 5 and 6).

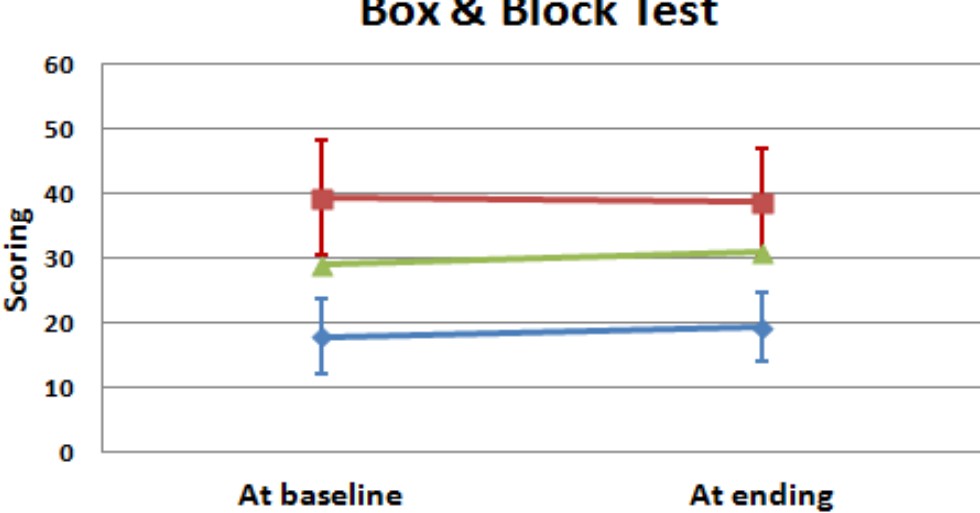

**Figure 6.** Graphical results in relation to baseline and end conditions for the Box and Block test. The tetraplegic group is expressed in blue, the paraplegic group in red and the case of a single patient in green.

It is worth showing the results obtained in a specific case of a patient with a spinal cord injury at the metameric level T2 (green color in Figures 5 and 6). This patient belongs to the paraplegic group. However, in terms of dexterity, his behavior was more similar to that of the tetraplegic group at baseline, whereas, at the end of the UE training, it was more similar to that of the paraplegic group. Figure 7 shows the trajectory followed by this participant in both assessments, at baseline and at the end, graphically reflecting the improvement experienced.

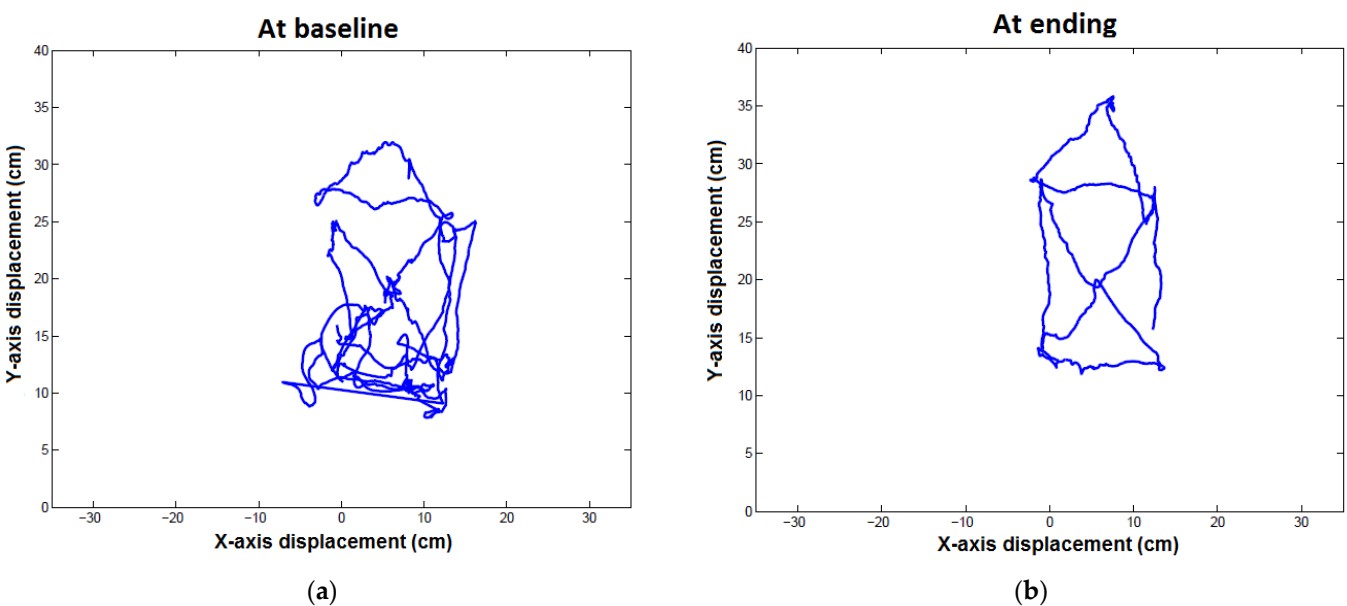

**Figure 7.** Hand path during the performance of the virtual application Explorer for a patient: (**a**) at baseline condition; (**b**) at ending the upper extremity training.

## 4. Discussion

The aim of the present work is to analyze, in a group of chronically spinal cord injured children, the smoothness and efficiency metrics after receiving UE training using the Robic humanoid robot as support. All our participants completed all the training sessions and maintained a sufficient level of attention during each of them to properly follow the robot's instructions under the supervision of the therapist.

We designed the training program based on some facts: (a) the hand is the main tool for manipulating the human environment, and to perform this function, it requires an efficient system of direction and transport provided by the shoulder, elbow and wrist [22]; (b) the central nervous system acts in a predictive fashion because it has "internal models", neural processes that can predict the effect of the interaction between different parts of the body and between the body and the environment [23]; (c) the central nervous system control muscles not one by one but in groups, in a task-specific way known as muscle synergy [24]. So, although our proposed robotic training program did not include specific exercises for the hands, nor did it include tasks aimed at improving their motor precision, we nevertheless expected, on the basis of the above concepts, to find some effect on hand movements related more to precision than to strength. And indeed, we found a tendency to shorten the hand trajectory with a reduction in the number of pikes, our most interesting findings, both related to the quality of upper limb motor control.

We did not expect to find changes in arm muscle strength (UEMS), but we also found no variations in Box and Block test performance attributable to the robotic platform intervention. The most plausible explanation could be that the intervention was of short duration (10 sessions) and of too low an intensity to produce noticeable changes in the selected muscle groups during the training period. Previous studies with children and adolescents with disabilities show increased strength, improved mental well-being and better overall function following the timing of strength training programs, which were carried out in the participants' homes three times a week for 6 weeks (a total of 18 prescribed sessions) [25]. Therefore, the significant differences found in muscle strength and trajectory between the tetraplegic and paraplegic groups are directly attributable to the spinal cord injury itself, all the more so when the selected participants had chronic lesions of several years' evolution and established sensorimotor impairments.

There is considerable evidence that voluntary exercise training improves the fitness of people with acute and chronic SCI, although the magnitude of the effect depends on the level and severity of SCI and the characteristics of the exercise program itself [26–29]. However, the level of endurance and work capacity in tetraplegics does not approach that of their paraplegic counterparts, with the level of SCI being the key determinant of the magnitude of improvement. In our case, it was the children with cervical spinal cord injuries who had the clearest benefits, most likely because the training protocol was not intense enough for those with paraplegia, as the program does not reach the minimum recommended level of activity—at least 30 min of moderate-intensity activity on 5 or more days per week, or at least 20 min of vigorous-intensity activity on 3 or more days per week [30].

Our data also show that even "suboptimal" activity patterns have beneficial effects on more precision-related motor tasks, such as those of the hands. In this sense, the findings in the length of the movement trajectory are particularly interesting. The movement execution using longer and more curved trajectories have been described in patients who suffer from sensory deafferentation in addition to motor loss, as is the case of spinal cord-injured patients [23]. As mentioned above, all participants in our study had chronic but incomplete injuries, with both sensory and motor remnants potentially trainable and varying degrees of disruption of upper limb motor synergies derived from their level of injury. The fact that the shortening of the hand movement trajectory occurs mainly, almost exclusively, in quadriplegic children is explained by the fact that it is these children who have upper limb functional deficits directly caused by SCI. With the exception of the T2 level SCI patient presented individually, all the paraplegic children in our sample had preserved

and functional all the circuits involved in the sensorimotor function of the arms so that when they followed the trajectories in the non-immersive virtual application based on the Leap Motion Controller (LMC), they had all the sensory resources to make the necessary feedforward corrections so that the hand movements were smooth and efficient. These kinds of improvements in quadriplegic patients would potentially enhance performance in other manipulative tasks of cognitive content, such as writing, so important in the age range of our selected patients. Probably the most promising aspect of our results is precisely that our data shows that these aspects can be trained so that the reorganization and optimization of motor synergies described in the motor learning processes of a given task can also be achieved by training the muscle groups involved in the synergies, in our case the arms [23,31,32].

With rehabilitation robotics, technologies have been introduced into the clinical environment to be integrated into the therapeutic process of patients. Specifically, assistive social robotics through the Robic robot provides guided therapy through a noncontact motor rehabilitation system that has been extensively evaluated and progressively enhanced [7]. The main limitation of the study is the size of the sample analyzed, this being the first experience of a Robic robot in a sample of pediatric patients with SCI. The improvements observed in patients related to the UE dexterity throughout the smoothness and efficiency metrics were measured by means of RehabHand software [20], which allows us to register kinematic data from the hand during the execution of the functional tasks.

For this reason, future works should address three different aspects: the optimization of the robotic platform so that it can provide real-time data on the subject's movement execution; the training protocols themselves, evaluating whether the effects that we have found in the present work are maintained or even improved by applying more demanding exercise protocols, in addition to analyzing how the reorganization of motor synergies occurs depending on the level and severity of the spinal cord injury and introducing a control group of SCI patients; and continue the research in the proposal of new metrics from kinematic hand data measured by means of the technologies applied in neurorehabilitation.

## 5. Conclusions

The significance of the results obtained is that patients with tetraplegia need to improve hand motor accuracy and dexterity prior to intervention, and with intervention, they train and improve. However, patients with paraplegia do not have, due to their injury, impaired manipulative dexterity of the UE. And with this intensive, short-term training, patients with paraplegia do not improve. With a longer workload, it is possible that this could also improve in this population. Thus, the findings of this research suggest further research along these lines, especially when most UE research in paraplegia tends to focus on the analysis of UE support strength for crutch use and wheelchair propulsion.

**Author Contributions:** Conceptualization, A.D.-G., A.G.-A., J.-C.P.-P. and E.L.-D.; methodology, M.S.-M., V.C.-H., A.D.-G., Y.P.-B., F.G.-M., J.-C.P.-P. and E.L.-D.; software, V.C.-H., A.D.-G. and J.-C.P.-P.; validation, A.D.-G., A.G.-A., J.-C.P.-P. and E.L.-D.; formal analysis, M.S.-M., V.C.-H., A.D.-G., Y.P.-B., J.-C.P.-P. and E.L.-D.; investigation, M.S.-M., V.C.-H., A.D.-G., Y.P.-B., F.G.-M., J.-C.P.-P. and E.L.-D.; resources, A.D.-G., A.G.-A., J.-C.P.-P. and E.L.-D.; data curation, M.S.-M., V.C.-H., A.D.-G., J.-C.P.-P. and E.L.-D.; writing—original draft preparation, M.S.-M., A.D.-G. and E.L.-D.; writing—review and editing, M.S.-M., V.C.-H., A.D.-G., Y.P.-B., F.G.-M., J.-C.P.-P. and E.L.-D.; visualization, A.D.-G., A.G.-A., J.-C.P.-P. and E.L.-D.; supervision, A.D.-G., J.-C.P.-P. and E.L.-D.; project administration, A.D.-G., J.-C.P.-P. and E.L.-D. All authors have read and agreed to the published version of the manuscript.

**Funding:** This research received no external funding.

**Informed Consent Statement:** Informed consent was obtained from all subjects involved in the study. Written informed consent has been obtained from the patient(s) to publish this paper.

**Data Availability Statement:** Not applicable.

**Conflicts of Interest:** The authors declare no conflict of interest.

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
