# Peer review of "Smoothness and Efficiency Metrics Behavior after an Upper Extremity Training with Robic Humanoid Robot in Paediatric Spinal Cord Injured Patients"

_applsci, doi:10.3390/app13084979_

Round 1
Reviewer 1 Report
This paper evaluates the clinical effect of extremity training with a humanoid robot on the behavior of pediatric spinal cord-injured patients. The authors set up a sensor system to track and record the patients’ hand trajectories and evaluated their smoothness and efficiency metrics. The results indicate a significantly longer trajectory for the patients with tetraplegia than those with paraplegia at baseline. Meanwhile, at the end of the training, there was no significant difference between the groups.
This is an interesting study that evaluated the effect of upper extremity training with a humanoid robot on the behavioral metrics for pediatric spinal cord injured patients. However, I have some reservations about accepting this paper in its current form for the following reasons.
My first question is whether the use of the humanoid robot, Robic, is properly justified in this study. Is there any specific reason for using this type of robot instead of other similar training such as video or human therapist? If not, the scope of this study can be seen as too narrow since it was conducted under a very special case. The authors could conduct a survey with the participants and compare the results to the one with other training method.
My second concern is about the missing information in the figures. On page 1, there is no caption but just a figure, which is possibly Figure 1. Figures 4 and 5 seem to be the captured image of the Excel missing description of what each color means.
Author Response
Comments and Suggestions for Authors
Reviewer: This paper evaluates the clinical effect of extremity training with a humanoid robot on the behavior of pediatric spinal cord-injured patients. The authors set up a sensor system to track and record the patients’ hand trajectories and evaluated their smoothness and efficiency metrics. The results indicate a significantly longer trajectory for the patients with tetraplegia than those with paraplegia at baseline. Meanwhile, at the end of the training, there was no significant difference between the groups.
Authors: The significance of this result is that patients with tetraplegia need to improve hand motor accuracy and dexterity prior to intervention and with intervention they train and improve. However, patients with paraplegia do not have, due to their injury, impaired manipulative dexterity of the upper extremity. And with this intensive, short-term training, patients with paraplegia do not improve. With a longer workload it is possible that this could also improve in this population. Thus, the findings of this research suggest further research along these lines, especially when most upper extremity research in paraplegia tends to focus on the analysis of upper extremity support strength for crutch use and wheelchair propulsion.
We provide two references related to studies centered in analyzing upper extremity strength:
- Gil-Agudo, A., Del Ama-Espinosa, A., Pérez-Rizo, E., Pérez-Nombela, S., & Crespo-Ruiz, B. (2010). Shoulder joint kinetics during wheelchair propulsion on a treadmill at two different speeds in spinal cord injury patients. Spinal cord, 48(4), 290–296. https://doi.org/10.1038/sc.2009.126
- Gil-Agudo, E. Pérez-Rizo, A. del Ama-Espinosa, AI. de La Peña-González, B. Crespo-Ruiz, A. Sánchez-Ramos. Assessment of walking on crutches in patients with incomplete spinal cord injury. Rehabilitación (Madr), 2009; 43(2):65-71. DOI: 10.1016/S0048-7120(09)70772-8
Reviewer: This is an interesting study that evaluated the effect of upper extremity training with a humanoid robot on the behavioral metrics for pediatric spinal cord injured patients. However, I have some reservations about accepting this paper in its current form for the following reasons.
My first question is whether the use of the humanoid robot, Robic, is properly justified in this study. Is there any specific reason for using this type of robot instead of other similar training such as video or human therapist?
Authors: To engage paediatric patients in training for longer periods of time, tools are needed that place the child in an environment somewhere between therapeutic and play. In this respect, humanoid robots are of great value. As an added value, these devices not only improve the environment, but also provide real-time data during the session, such as feedback to the child based on their motor performance. In the particular case of children with spinal cord injury who need a high amount of rehabilitation treatment throughout their childhood, it is essential to carry out work such as this one to demonstrate that training with humanoid robotic platforms is possible, can be carried out and has evident motor effects.
Two new references have been added in the Introduction section:
- Guneysu, A.; Arnrich, B. Socially assistive child-robot interaction in physical exercise coaching. In 2017 26th IEEE international symposium on robot and human interactive communication (RO-MAN) 2017, August, (pp. 670-675). IEEE.
- Casas, J.; Irfan, B.; Senft, E.; Gutiérrez, L.; Rincon-Roncancio, M.; Munera, M.; ... & Cifuentes, C. A. Social assistive robot for cardiac rehabilitation: A pilot study with patients with angioplasty. In Companion of the 2018 ACM/IEEE International Conference on Human-Robot Interaction 2018, March (pp. 79-80).
Reviewer: If not, the scope of this study can be seen as too narrow since it was conducted under a very special case. The authors could conduct a survey with the participants and compare the results to the one with other training method.
Authors: The reviewer is right that in order to get the most out of the different types of robotic devices, a survey of potential users and their relatives would be of interest. However, in our opinion this does not replace the need to evaluate the effect that each of the robotic therapies has on patient training. This work is one of those needed in this regard. We thank the reviewer for making us think about users.
Reviewer: My second concern is about the missing information in the figures. On page 1, there is no caption but just a figure, which is possibly Figure 1. Figures 4 and 5 seem to be the captured image of the Excel missing description of what each color means.
Authors: Regarding Figure 1, the legend is at the top of the next page. This is an editing problem that will be solved when the final version is edited.
Figure 4 and Figure 5. The presentation of the figures was adapted to the template provided by the journal, so the explanations were placed in the caption of the figure caption in text mode.

Reviewer 2 Report
This is valuable work. The article deals with a current problem related to the rehabilitation of the upper limbs using a humanoid. Here are some comments to improve this paper,
1. The abstract should be started with the problem statement.
2. It is better to mention the contribution of the paper in bullet points in the introduction.
3. The contents of the paper should be mentioned at the end of the introduction
4. There is a lack of methodology in this paper. I cannot see how the training process works.
5. Please use the .eps format for the graphs.
6. There is no conclusion section in this paper
Author Response
Comments and Suggestions for Authors
Reviewer: This is valuable work. The article deals with a current problem related to the rehabilitation of the upper limbs using a humanoid. Here are some comments to improve this paper,
Authors: The authors thank the comments and suggestions of the reviewer for improving the paper
Reviewer: 1. The abstract should be started with the problem statement.
Authors: The abstract have been modified according to reviewer’s suggestion.
Reviewer: 2. It is better to mention the contribution of the paper in bullet points in the introduction. 3. The contents of the paper should be mentioned at the end of the introduction
Authors: The content of the paper has been described at the end of the introduction section in bullet points, following to reviewer’s suggestion.
Reviewer: 4. There is a lack of methodology in this paper. I cannot see how the training
process works.
Authors: We agree to the reviewer that the training process is not clearly detailed, so we have included a new Figure explaining the content of an experimental session in a graphical way.
Reviewer: 5. Please use the .eps format for the graphs.
Authors: Thanks for the comment. We have tried to present all the graphs in this format .eps. Editors can use the more convenient format.
Reviewer: 6. There is no conclusion section in this paper
Authors: Sorry for the missing. The conclusion section has been included.
